# Strong volcanic-induced climatic shocks on historical Moselle wine production

Fredrik Charpentier Ljungqvist[1,2,3], Bo Christiansen[4], Lea Schneider[5,6], and Peter Thejll[4]

[1]Department of History, Stockholm University, 106 91 Stockholm, Sweden
[2]Bolin Centre for Climate Research, Stockholm University, 106 91 Stockholm, Sweden
[3]Swedish Collegium for Advanced Study, Linneanum, Villavägen 6c, 752 38 Uppsala, Sweden
[4]Danish Meteorological Institute, Sankt Kjelds Plads 11, 2100 Copenhagen Ø, Denmark
[5]Department of Geography, Justus-Liebig-University, 35390 Giessen, Germany
[6]Center for international Development and Environmental Research, Justus-Liebig-University, 35390 Giessen, Germany

**Correspondence:** Fredrik Charpentier Ljungqvist (fredrik.c.l@historia.su.se)

**Abstract.** In central and southern Europe, grapevine is a climate-sensitive agricultural product of great economic importance, both in historical times and today. We systematically investigated the climatic impact, focusing on volcanic-forced abrupt cooling, on two long annual records of wine production quantity (spanning 1444–1786) from the Moselle Valley in present-day Luxembourg, close to the northern limit of viticulture in Europe. We present a consistent picture of the impact of volcanic eruptions on wine production through climate. To this end, we applied superposed epoch analysis – an appropriate method for detecting episodic signals in non-stationary time-series – in combination with a bootstrap procedure to estimate the statistical significance. We also assessed the long-term relationship between different annual and seasonal climate parameters and wine production in the Moselle Valley. Robust and highly significant wine production declines occurred in the years immediately following major volcanic events. Warmer, and to a lesser extent drier, climate condition had a moderately strong, but persistent, positive effect on wine production. We also find a volcanic cooling signature in spring and summer in temperature reconstructions. However, the detected volcanic signature in the Moselle Valley wine production is considerably stronger than the one found for Central Europe in tree-ring-based reconstructions and is instead more akin to the strong volcanic signature present in Fennoscandian tree-ring series. On the basis of our findings, we encourage further compilation, publication, and analyses of additional wine production series containing unique biological and climatic information.

## 1 Introduction

The climatic response to volcanic forcing during the past few millennia has been studied intensively from local to global scales, typically using tree-ring data from strongly temperature-limited growth environments. The impacts on agricultural productivity, including on viticulture, in less extreme climatic settings remain far less well understood. The study of long time-series of agricultural productivity, like grapevine harvest quantities, offers insights into the effects of such abrupt cooling on biological records with properties possibly differing from those of tree-ring series. This, in turn, may shed new light on still unresolved

questions about the effects volcanic-induced cooling has on tree growth, in addition to gaining a better understanding of the role of major volcanic events for past agriculture.

Grapevine (henceforth 'vine') is a highly climate-sensitive crop whose phenology is comparatively well understood (Jones et al., 2005; Bock et al., 2013; de Cortazar Atauri et al., 2017; Drappier et al., 2019; Morales-Castilla et al., 2020; Droulia and Charalampopoulos, 2022). The biological requirements of both modern and historical European viticulture have been documented in detail (Lachiver, 1988; Mullins et al., 1992; Guerreau, 1995). An increasingly warmer climate, with longer growing seasons, benefits viticulture (vine growing) near its latitudinal and altitudinal limits (Malheiro et al., 2010; Guiot et al., 2023), whereas climatic warming at the same time can have adverse effects on viticulture in regions with a hot and dry Mediterranean climate (Cook and Wolkovich, 2016; Venios et al., 2020; Andrade et al., 2024; van Leeuwen et al., 2024). Grapes were the most important cash crop, of an economic importance only second to grain, in much of Europe during the medieval and early modern periods, and vine is and was more climate-sensitive than grain (Pfister and Wanner, 2021). Relationships between different parameters associated with wine production and summer temperature in late medieval and early modern Europe have been previously described in the literature (Ladurie Le Roy and Baulant, 1980; Le Roy Ladurie et al., 2006; Labbé et al., 2019; Pfister et al., 2024). In Central Europe, the quantity of the grape harvest was positively correlated to June–July temperature, the quality to July–September (and especially August) temperature, and the harvest dates to April–July/August temperature (Pfister, 1981; Pfister et al., 2024). Viticulture in Central Europe is, and was, especially sensitive to the effect of spring frost (Meier et al., 2018). The strong cooling during the climax of the Little Ice Age *c.* 1570–1710 (Wanner et al., 2022) made viticulture non-viable in parts of Central Europe – with considerable economic and social consequences (Landsteiner, 1999).

Large, but more short-lived, reductions in vine harvests – and thus wine production declines – could be triggered by sharp volcanic-induced cooling. Major volcanic eruptions have a significant impact on climate (Robock, 2000; Oppenheimer, 2011) through a negative radiative forcing related to an increased atmospheric loading of aerosols and dust, and can induce significant global to hemispheric scale cooling for one to several years (Stoffel et al., 2015; Sigl et al., 2015; Burke et al., 2023). The climate effect of an eruption does not only depend on its magnitude, but also on factors such as the geographical position of the volcano and the season of the eruption (Guillet et al., 2017). Quantitative estimates of volcanic-induced cooling during pre-industrial times have mostly been derived from large-scale temperature reconstructions based on tree-ring data (Schneider et al., 2015; Stoffel et al., 2015; Wilson et al., 2016; Anchukaitis et al., 2017; Büntgen et al., 2021). In altitudinal and latitudinal tree-line ecotones, tree growth is particularly sensitive to summer temperature (Körner, 2021) and short-term climatic perturbations are archived in anomalies of annual ring width or wood density (Esper et al., 2016; Hartl-Meier et al., 2017). The precise annual dating of tree-ring chronologies allows for a clear attribution of temperature deviations to volcanic eruptions (Anchukaitis et al., 2012; D'Arrigo et al., 2013; Esper et al., 2013a; Schneider et al., 2017).

The adverse impacts on agricultural productivity and, hence, on human society of volcanic-induced cooling in pre-industrial times have been well-documented in recent scholarship (D'Arrigo et al., 2020; Guillet et al., 2020; Huhtamaa et al., 2022; Stoffel et al., 2022; White et al., 2022; Martin et al., 2023). This is, in particular, the case at the northern edge of grain agriculture in Europe (Ljungqvist et al., 2021, 2024). The effects of volcanic-induced cooling on viticulture in pre-industrial

Europe remains less well understood. However, it has been demonstrated that larger, especially tropical, volcanic eruptions resulted in significantly later grape harvest dates for one to two years after the volcanic event (Meier et al., 2007; Guillet et al., 2017; Pfister and Wanner, 2021). The effects of individual major volcanic forcing events on wine production have also been studied. For example, Brönnimann and Krämer (2016) studied the effects of the 1815 Tambora volcano eruption, followed by the cold summer of 1816 in central Europe. The extent to which the cooling was due to volcanic forcing was evaluated, as well as its biophysical effects (e.g., low harvest yields) and subsequent societal consequences.

Numerous records, not least from France and Switzerland, exist of grape harvest dates (Yiou et al., 2012). They have been proven to be strongly related to spring and summer temperatures, and have been utilised in numerous documentary-based temperature reconstructions (e.g., Chuine et al., 2004; Kiss et al., 2011; Možný et al., 2016a; Labbé et al., 2019), and also drought reconstructions (Možný et al., 2016b), despite possible biases in the long-term trends (Garcia de Cortazar-Atauri et al., 2010; Garnier et al., 2011). Grape harvest dates have been proven to be particularly skillful in capturing extremely cold years (Maurer et al., 2011), such as those frequently following volcanic forcing events (Sigl et al., 2015), although the skill of capturing extremely warm years are less good (Keenan, 2007). Likewise, a number of long wine price series exist from different portions of Europe (Allen and Unger, 2019), but the possible climate signal embedded in the wine price series remains relatively unexplored. In contrast to grape harvest dates and wine price series, long and continuous series of wine production (or grapevine harvest quantity or quality) are rare for medieval and early modern Europe (see, though, the recently published ones by Pfister et al., 2024). However, two long annual wine production quantity records, covering 1444–1786 with gaps, were already published by Yante (1985). They derive from Grevenmacher and Remich in the Moselle Valley, in present-day Luxembourg, close to the latitudinal limit of European viticulture (Fig. 1a–b).

In this article, we systematically investigate the impact from climate, with particular focus on the influence from volcanic forcing, on the two above-mentioned long wine production quantity series. We will first assess the effects of volcanic forced cooling on wine production using superposed epoch analysis (SEA) appropriate for studying episodic events in non-stationary time-series. Then, we will evaluate the effects of volcanic forcing on Moselle Valley pre-industrial climate and explore whether the wine production series contribute information useful for understanding volcanic-induced cooling in the region between the Alps and Scandinavia in comparison to temperature-sensitive tree-ring data. Finally, we will investigate both the short- and long-term relationship between different annual and seasonal climate parameters and wine production quantities in order to understand and constrain the effects of climate on wine production variability. Through utilising bootstrap and phase-scrambling-based techniques to estimate significance, we intend to pay particular attention to issues related to statistical significance.

We emphasise that this article is of a statistical nature. Our aim is to investigate the statistical relationships between volcanic forcing, climate and wine production in the Moselle Valley region. The rest of article is organised as follows: After presenting our data and methods (Section 2), we start with investigating the influence of volcanic eruptions on wine production (subsection 3.1), followed by the influence of volcanic eruptions on climate (subsection 3.2), and ending with the influence of climate extremes on the wine production (subsection 3.3). In Section 4 we discuss the interpretations and implications of the results and their associated uncertainties, starting with the volcanic and climatic signals embedded in the wine production series (sub-

 section 4.1) followed by a comparative discussion about the volcanic forcing effects on wine production in relation to tree-ring growth (subsection 4.2). We conclude the article with a short conclusion and outlook (Section 5).

## 2 Materials and methods

In this section we start with describing the wine production data (Section 2.1), followed by the volcanic forcing dataset (Section 2.2), and then the (palaeo)climate series (Section 2.3). All the datasets are summarised in Table 2. We end the section with
95 describing the various statistical methods applied (Section 2.4).

### 2.1 The Moselle Valley wine production series

The two annual wine production quantity series from Grevenmacher and Remich in the Moselle Valley (49.5°N, 6.35°E) of present-day Luxembourg are based on official records of a tax of one-ninth levied on the wine production. They cover the 1444–1786 period, though with a gap between 1684–1741 (except for the years 1698–1701) in connection with the French
occupation of Luxembourg (Yante, 1985). It is unfortunate that much of the cold period coinciding with Maunder Minimum of solar activity ($c$. 1645–1715; Eddy, 1976; Steinhilber et al., 2009) is missing in the Moselle Valley wine production data. However, the last three decades of the 16th century were at least as cold as any part of the Maunder Minimum (Pfister and Wanner, 2021; Wanner et al., 2022). Thus, we still capture the full range of temperature variability.

While the tax was originally levied, when introduced in the 13th and 14th centuries, on every ninth basket of grapevines
harvested, it soon instead became levied on every ninth barrel of wine produced. However, the number of barrels of wine produced closely followed the number of baskets of grapevines harvest. Thus, the number of barrels of wine produced mimics the quantity of the vine harvest. It is uncertain to what extent changes over time in tax exemptions affected the wine quantity at inter-annual and longer time-scales. Furthermore, the tax appears in fact to, at times, have been both higher and lower than one-ninth of the wine production. The degree of fraud and resistance to taxation might also have varied over time (see also Pfister,
1984). Sufficient to say is that the long-term trend in the Grevenmacher and Remich series are not reliable. The source-critical problems associated with tithe records to estimate actual agricultural productivity variations have been extensively treated in the agrarian history literature (e.g., Kain, 1979; Le Roy Ladurie and Goy, 1982; Leijonhufvud, 2001; Skoglund, 2023). This research on tithes as a historical source for agricultural productivity has shown that long-term trends contain larger biases than short-term variability. Grevenmacher shows a long-term rather stable, or slightly decreasing, production trend, whereas
Remich shows a long-term production increase (Fig. 1a). Nevertheless, the correlation coefficients between the Grevenmacher and Remich series for linearly detrended and 11-year high-pass filtered data, respectively, are $r = 0.55$ and $r = 0.62$. Both correlations are significant at a $p = 0.01$ level (two-tailed test).

Modern observations show that temperature is the single most important climatic factor for viticulture in the Moselle region (Urhausen et al., 2011). The soil in Grevenmacher is limestone and in Remich is alluvial. Grain is the main crop but
terraces of vines are widespread too (Yante, 1985). The annual mean temperature during the twentieth century was around 10°C with around 650 mm annual precipitation and slightly over 1400 hours of sunshine. Years with less than 500 mm pre-

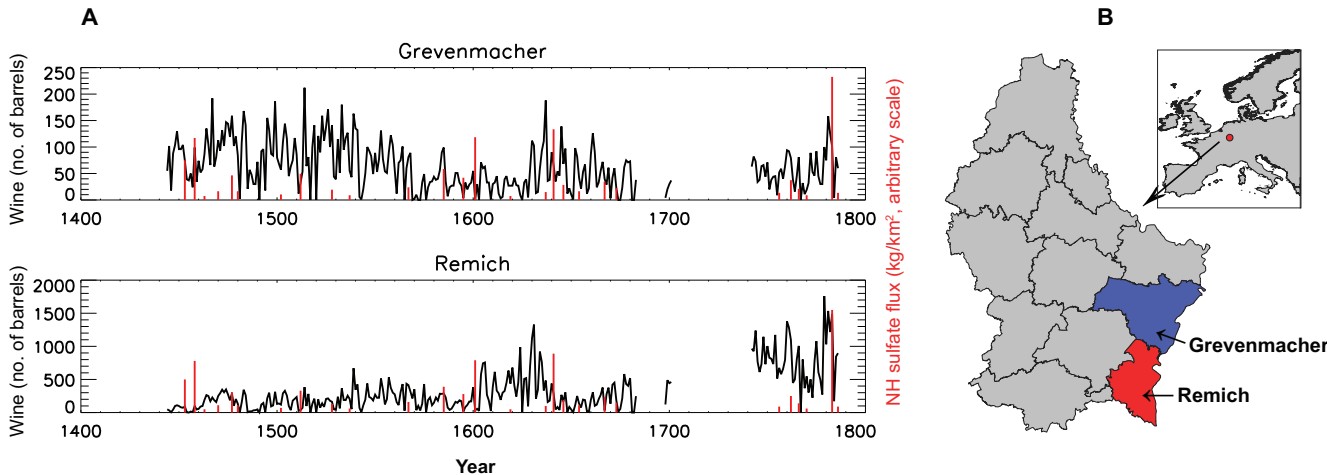

**Figure 1. A** Annual wine production in number of barrels for each series (*in black*) and volcanic sulfate flux derived from Greenland sulfate deposition in kg/km$^2$ (*in red*). **B** Location of the two sites with wine data from 1444–1786 (with gaps) in the Moselle Valley of Luxembourg.

cipitation tend to produce the best wine. Spring frost is a known problem for viticulture in the region (Meier et al., 2018). The Moselle Valley was during the early modern period a rather marginal viticulture area (about 90% of the wine was white) with often relatively low wine quality, with the wine of Remich considered superior to that of Grevenmacher. Consequently, the vine 125 was grown to much larger extent in Remich than in Grevenmacher (Yante, 1985).

## 2.2 Volcanic forcing data

We employ the volcanic stratospheric sulfur injection data from Toohey and Sigl (2017) (lists provided by M. Sigl, priv. comm., 2022), an updated version of Sigl et al. (2015), that is utilised for the recommended forcing in state-of-the-art PMIP4 'past1000' climate model simulations (Jungclaus et al., 2017). This dataset provides forcing year, not eruption year (which is 130 sometimes unknown), and depending on the seasonality of the eruption the peak forcing year can be the same year as eruption year or the following year. Toohey and Sigl (2017) used composite ice-core data from Greenland and Antarctica, separately, to generate a robust list of dates of volcanic eruptions. We only use the dates for forcing events with aerosol loading in the Northern Hemisphere (NH) (flux is provided as the deposited sulfate found in the ice cores given in kg/km$^2$). We restrict our list to those eruptions, since it is likely that even strong Southern Hemisphere extra-tropical eruptions do not contribute much 135 to an aerosol loading in the NH.

For use in the superposed epoch analysis, we identify key years with NH volcanic forcing of various strengths. In the Toohey and Sigl (2017) NH flux series, for the period we use in the analysis, there are 34 (27) eruptions listed with strengths above 0 kg/km$^2$ (the smallest eruption in the series is near 3.2 kg/km$^2$); there are 28 (22) eruptions stronger than 5 kg/km$^2$; 16 (12) stronger than 10 kg/km$^2$; 11 (7) stronger than 15 kg/km$^2$, and 9 (6) stronger than 20 kg/km$^2$. Here, the numbers in parentheses 140 give the count of eruptions that also cover complete wine data, in the sense that we only include eruptions not less than 5

**Table 1.** Year and flux (in kg/km$^2$) of the volcanic forcing events 1444–1786 based on Toohey and Sigl (2017). We include the volcanic events occurring 1684–1741 when we have a gap in the wine production series because the events are used for the SEA analysis based on climate extremes.

| Year | NH flux [kg/km$^2$] | Year | NH flux [kg/km$^2$] |
|------|---------------------|------|---------------------|
| 1453 | 24.80 | 1646 | 12.70 |
| 1458 | 35.30 | 1654 | 6.20 |
| 1463 | 3.20 | 1667 | 18.30 |
| 1470 | 7.30 | 1673 | 5.90 |
| 1477 | 27.00 | 1693 | 6.96 |
| 1480 | 8.70 | 1695 | 24.72 |
| 1502 | 5.30 | 1707 | 5.71 |
| 1510 | 12.10 | 1720 | 3.77 |
| 1528 | 5.70 | 1721 | 4.28 |
| 1537 | 4.50 | 1729 | 25.37 |
| 1554 | 5.20 | 1739 | 18.09 |
| 1567 | 13.20 | 1755 | 6.20 |
| 1585 | 23.30 | 1762 | 9.30 |
| 1595 | 11.80 | 1766 | 13.20 |
| 1600 | 38.10 | 1770 | 3.70 |
| 1637 | 6.80 | 1783 | 109.50 |
| 1640 | 41.10 | 1786 | 4.40 |

years from a wine data gap. In our analyses, we use all the eruptions, as recent results suggest that weak eruptions can have significant climate effects (Chim et al., 2023). However, we also tested the effects of using only eruptions larger than the thresholds mentioned above. The eruption years included are listed in Table 1. In addition, tested the sensitivity of the SEA by progressively excluding the strongest volcanic eruptions.

**2.3   Climate data**

Instrumental climate data have only a short period of overlap with the Grevenmacher and Remich wine production series. The closest instrumental temperature station to Luxembourg in the homogenised European Climate Assessment (ECA) temperature dataset (Squintu et al., 2019) is the monthly-averaged data for De Bilt in The Netherlands (van Engelen and Geurts, 1985) at a distance of about 300 km. Data from De Bilt overlaps with the wine production series from 1742 to 1786, covering only a
150 period of 45 years. The short overlap with the De Bilt instrumental record limits the reliability and confidence in results derived from it.

We therefore employ reconstructed series of temperature and precipitation, as summarised in Table 2. We use seasonally resolved gridded reconstructed temperature data from Luterbacher et al. (2004) and Xoplaki et al. (2005) (henceforth only cited as Luterbacher et al. (2004)) and seasonally resolved gridded reconstructed precipitation from Pauling et al. (2006)
back to 1500 (Table 2). Both the Luterbacher et al. (2004) and Pauling et al. (2006) reconstructions include instrumental data during the 18th century. Data are extracted from the local grid-cell(s) covering Luxembourg. In addition, we use the seasonal resolved reconstructed temperature data from Dobrovolný et al. (2010), which is available as a Central European average for the same period. For the full period since 1444 we use the June–August temperature reconstruction by Luterbacher et al. (2016), as updated by Ljungqvist et al. (2019), based mainly on tree-ring data but also containing the documentary-based June–
August season data from above-mentioned Dobrovolný et al. (2010) reconstruction. Furthermore, we employ for the full period the solely tree-ring-based NTREND gridded field reconstruction (Anchukaitis et al., 2017) of May–August temperature. For comparisons against volcanic-induced cooling in the Alpine region, we use the June–September temperature reconstruction from the Lötschental, Switzerland, which is derived from a strong temperature signal in maximum latewood density tree-ring series (Büntgen et al., 2006). June–August soil moisture (drought) conditions for the entire period are obtained from the tree-
ring width based gridded Old World Drought Atlas (Cook et al., 2015) providing self-calibrated Palmer Drought Severity Index (scPDSI) values (van der Schrier et al., 2011).

## 2.4   Statistical methods

The episodic nature of the volcanic eruptions is an advantage when analysing their effects on variables with unknown, or even dubious, low-frequency variations and trends, such as wine production quantities. We simply compare the strength of the mean
superposed standardised signal of the considered time-series (wine production or climate conditions) in the years immediately following the eruptions with the mean superposed standardised signal in the years before (five years). Using superposed epoch analysis (SEA; Chree, 1913, 1914; von Storch and Zwiers, 1999; Rao et al., 2019), we calculate the average of this difference across $K$ eruptions, that are covered by the considered time-series – so the number of eruptions might vary slightly for different time-series – and stronger than the chosen threshold. We standardised the time-series to zero mean and unit variance before
conducting the SEA. The significance is estimated by a bootstrap procedure, assuming only that the eruptions are independent. If we have $K$ eruptions, we pick $K$ random years in the whole period under consideration and calculate the superposed epochs from these years. Note, that the years with actual volcanic eruptions should be selected with the same probability as any other year. Otherwise, a bias would be introduced. We do this 3000 times to get a distribution of the SEA under the null-hypothesis that volcanic eruptions have no impact on the considered time-series (e.g., wine production). Comparing the original SEA
with this distribution, we can calculate the $p$-value as the probability of having a lower SEA than the observed. We report significance in the SEA at both the $p = 0.05$ and $p = 0.01$ significance levels. Furthermore, our results are robust to changes in the details of the analysis, such as in the length of the epochs or whether the average is calculated over the whole epoch and not just the half preceding the eruptions. The SEA results are based on time-series standardised to unit variance over the whole period. However, similar results are obtained when the standardisation is performed individually over running-windows
(corresponding to variance equalisation).

**Table 2.** Wine production data, (palaeo)climate and volcanic forcing data series used in this study with information of parameters, period covered, grid-cell (centre coordinates), season, data type, AR1 (autoregressive coefficient at lag 1), and data source.

| Data-sets | Period | Grid-cell | Season | Data type(s) | AR1 | Source/reference |
|---|---|---|---|---|---|---|
| *Wine production data* | | | | | | |
| Grevenmacher | 1444–1786 | 49.5°N, 6.35°E | Annual | Documentary | 0.41 | Yante (1985) |
| Remich | 1444–1786 | 49.5°N, 6.35°E | Annual | Documentary | 0.62 | Yante (1985) |
| | | | | | | |
| *Temperature data* | | | | | | |
| Moselle temperature | 1500– | 49.5°N, 6.5°E | Seasonal | Multi-proxy/instr. | 0.04[a] | Luterbacher et al. (2004); Xoplaki et al. (2005) |
| Central Europe | 1500– | Not gridded | Seasonal | Documentary | 0.06[a] | Dobrovolný et al. (2010) |
| EuroMed2k (updated)[b] | Full | 47.5°N, 7.5°E | Jun–Aug | Tree-ring/doc. | 0.67 | Ljungqvist et al. (2019) |
| NTREND | Full | 47.5°N, 7.5°E | May–Aug | Tree-ring | 0.45 | Anchukaitis et al. (2017) |
| Lötschental, Switzerland | Full | 46.42°N, 7.83°E | Jun–Sept | Tree-ring | 0.55 | Büntgen et al. (2006) |
| | | | | | | |
| *Hydroclimate data* | | | | | | |
| Moselle precipitation | 1500– | 49.5°N, 6.5°E | Seasonal | Multi-proxy/instr. | 0.07[a] | Pauling et al. (2006) |
| scPDSI | Full | 49.5°N, 6.5°E | JJA | Tree-ring | 0.39 | Cook et al. (2015) |
| | | | | | | |
| *Volcanic forcing data* | | | | | | |
| NH flux | Full | Not gridded | Annual | Ice-core | — | Toohey and Sigl (2017) |

[a] Mean AR1 of the four different seasonal windows (December–February, March–May, June–August, and September–November).

[b] Updated by Ljungqvist et al. (2019) from the first EuroMed2k reconstruction version published by Luterbacher et al. (2016).

To further investigate the relationship between climate and wine production quantities, we calculated the Pearson correlation coefficient ($r$) between the wine production series and the various (palaeo)climate reconstructions. The significance of correlations were calculated using $p = 0.05$ with a phase-scrambling test, which is more conservative than the common Student's $t$-test (for details, see Ljungqvist et al., 2022, 2023). To emphasise inter-annual variability, we employed 11-year high-pass filtering of the data. For studying low-frequency variability, we likewise employed an 11-year low-pass filter. This emphasises variability at decadal scales, while inter-annual fluctuations and noise are suppressed. The choice of 11 years as the filter cut-off length is chosen to match the time-scale of the SEA. Although we have used a simple 11-year smoothing filter, other more complicated filters (e.g., Gaussian filters) provided almost identical results. During an exploratory data analysis phase, we also compared the use of the parametric Pearson correlation coefficient with the non-parametric Kendall's and Spearman correlation coefficients, finding that they resulted in similar correlation patterns.

# 3 Results

In this section we study the influence of volcanic eruptions on wine production (subsection 3.1), the influence of volcanic eruptions on climate (subsection 3.2), and the influence of climate extremes on wine production (subsection 3.3). We will mainly

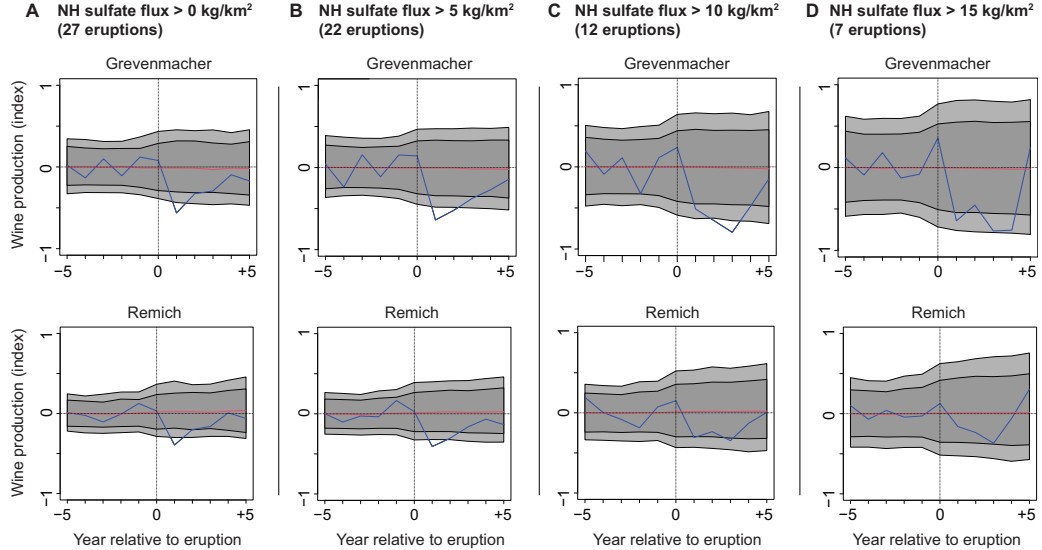

**Figure 2.** Superposed epoch analysis (SEA) of volcanic forcing, after NH sulfate flux $kg/km^2$ size, on standardised wine production quantity for Grevenmacher, Remich. The grey bands indicate $p = 0.05$ (light grey) and $p = 0.01$ (dark grey) significance levels, calculated by a bootstrap method (randomly selected key years).

focus on the episodic connection using SEA analysis, while we will additionally use correlation analysis in subsection 3.3.
Together, this will provide evidence for a volcanic and climatic influence on the wine production quantity, mainly through spring and summer temperatures.

### 3.1  Effects of volcanic forcing on Moselle Valley wine production

A main focus of this study is the effects of volcanic-forced abrupt cooling on the Moselle Valley wine production. Superposed epoch analysis (SEA) reveals a strong decrease in wine production the year following volcanic forcing events for both
Grevenmacher and Remich. The SEA signal itself is shown in Fig. 2 with the blue curves, while the dark and light shaded bands indicate the $p = 0.01$ and $p = 0.05$ significance levels, respectively. This volcanic-related production decrease pattern is similar regardless of whether all volcanic events are included or only those exceeding certain thresholds for NH forcing flux (as demonstrated in the different panels in Fig. 2). In addition, progressively excluding the strongest volcanic eruptions one by one had only a minor impact on the results (not shown). Conclusions were unchanged.
In general, we find a strong, significant, and consistent decrease in wine production in the years following volcanic forcing events for both the Grevenmacher and Remich series. For all thresholds the signal (blue curve in Fig. 2) is about half a standard deviation and the $p$-value is below 0.05 and often below 0.01 (shaded regions in Fig. 2). Slightly smaller signals are found for Remich compared to Grevenmacher. Including all eruptions, we find the largest signal in the first year after the eruptions. The (taxed) wine production was even zero, or close to it, in the first year following certain major volcanic events: 1481, 1601,

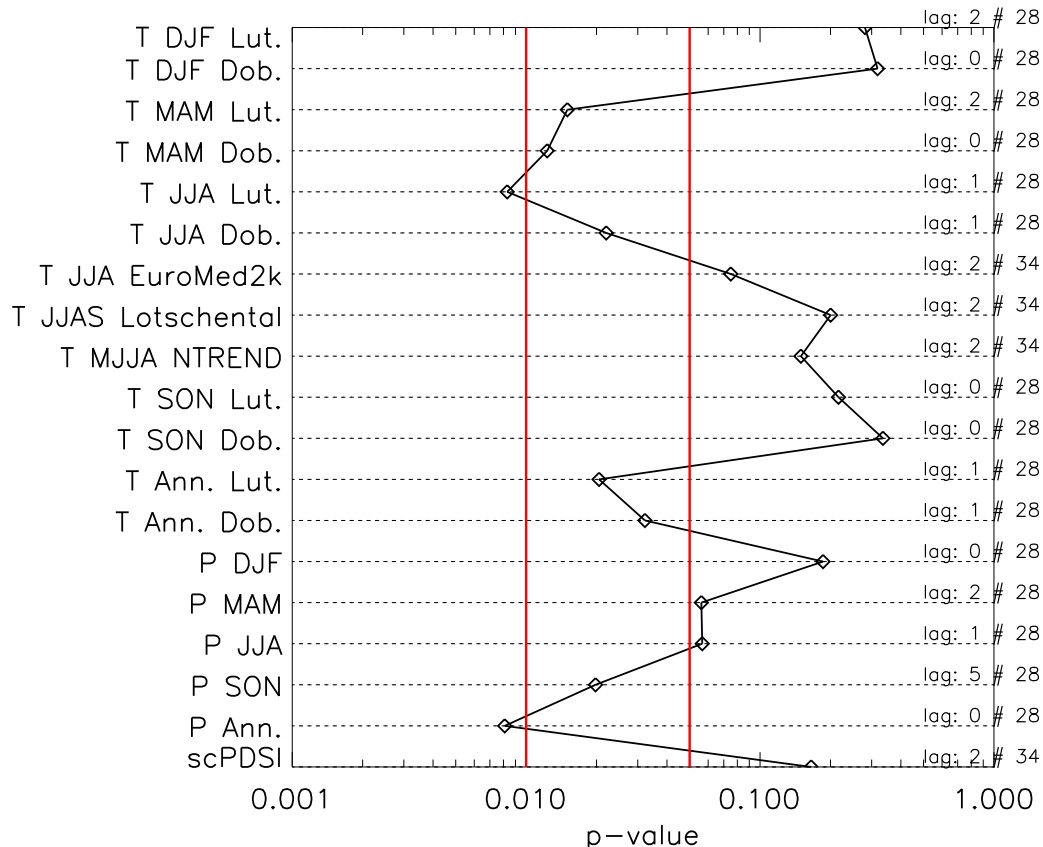

**Figure 3.** For each climate series in Table 2, a SEA based on the years of the volcanic forcing event has been performed. The lowest *p*-value (two-tailed test) in the year of the volcanic forcing event and the following 5 years are shown. The x-axis is logarithmic, and *p*-values of 0.01 and 0.05 are shown with the red vertical lines. The *p*-values are calculated using the bootstrap method. The significant signals in the temperature/precipitation series all correspond to colder/wetter conditions. To the right is shown the lag where the lowest *p*-value is found. Volcanic forcing events of all strengths are included, the exact number is dependent on data availability and is also shown.

1674, 1767, and 1784. However, when increasing the threshold and only including stronger (and fewer) eruptions, we see a tendency for the strongest wine production decrease to be delayed. For the largest eruptions, the strongest signals appear 3–4 years after the eruptions.

The bands indicating the significance levels expand with increasing threshold due to the fewer eruptions included (Fig. 2). When only including larger volcanic events, exceeding an estimated NH forcing flux of >10, a wine production decrease that is significant at the $p = 0.01$ level is only found the third year after the volcanic event – and only for Grevenmacher. Weaker, less significant, decreases are also detected the first and second years after the volcanic event, with the Remich series showing a broad decrease for years 1–3 after such strong eruptions.

## 3.2 Effects of volcanic forcing on Moselle Valley pre-industrial climate

We have performed SEA from volcanic forcing events on temperature and precipitation series (local grid-cells when applicable) from the Moselle Valley (see Table 2). Figure 3 summarises the strengths of the connection by reporting the smallest $p$-values (two-tailed test) taken over years 0–5, i.e., from the year of the eruptions and the following 5 years. These $p$-values basically reflect the strength of the SEA signals.

We find that significant connections for the temperature series mainly in spring and summer. For example, $p$-values below 0.02 are found for the temperature series by Dobrovolný et al. (2010) and Luterbacher et al. (2004) only for the March–May and June–August seasons. For temperature in spring and summer, the lowest $p$-values are found for lag 0 or 1 year (numbers shown to the right in Fig. 3). The significant signals in the temperature series all correspond to coolings. Similar results are obtained using only eruptions larger than 5 $kg/km^2$, while the situation for higher thresholds (and correspondingly fewer eruptions) is more muddled. For precipitation, significant signals corresponding to wetter conditions are found in annual and September–November reconstructions.

We emphasise the weak and insignificant volcanic signatures in the tree-ring based temperature reconstructions (EuroMed2k, NTREND, and Lötschental) for the Central Europe region. This less distinct volcanic signature stands in sharp contrast to the strong and clear wine production quantity reduction after volcanic forcing events, that reaches significance well below the $p = 0.01$ level for several years even when including minor forcing events (Fig. 2).

## 3.3 Effects of climate on Moselle Valley wine production

We have identified the 20 most extreme years (for each of the four seasons) for a selection of the climate series in Table 2. Thus, for temperature series we find the 20 most warm and cold years and for precipitation the 20 most wet and dry years. These extremes are identified from the raw series without detrending. We then conducted SEA to identify the responses in the two wine production series to these four kinds of extremes for each season. The results are shown in Table 3 which reports the $p$-values and the signs of the significant responses.

We find that the spring (MAM) stands out from the other seasons with significant responses for all four kinds of extremes. We also note that consistent directions of the responses are found; increasing wine production for warm/dry conditions and decreasing production for cold/wet conditions. The climate series from Luterbacher et al. (2004) and Pauling et al. (2006) show consistent results. Furthermore, we find almost identical results for the Remich and for Grevenmacher wine series. For the other seasons, fewer significant and less consistent responses are found, indicating that the significance in these seasons might partly be due to chance.

Through correlation analyses we explored the long-term relationships between wine production and seasonal and annual temperature, precipitation and soil moisture. We started the analyses with De Bilt instrumental series. The correlation between the De Bilt instrumental series and the Grevenmacher series, over their 45-year period of overlap 1742–1786, is $r = 0.46$ using 11-year high-pass filtered data for JJA temperature for both. For Remich the result is $r = 0.29$ (just below significance). Furthermore, annual mean temperature for De Bilt shows significant positive correlations with Grevenmacher and Remich of $r$

**Table 3.** The *p*-values for the SEA performed using as key years the 20 warmest/coldest/wettest/driest years, respectively, in the seasonal series of the Luterbacher et al. (2004) temperature reconstruction and the Pauling et al. (2006) precipitation reconstruction on the Remich and Grevenmacher wine production series. The *p*-values (two-tailed test) were calculated using repeated scrambling of years (bootstrap), see text for details, and *p*-values below 0.05 are typeset in bold. The lowest *p*-value (two-tailed test) in the year of the eruptions and the following 5 years are shown in the table. (+) and (–) designate the direction of the effect on the wine series, where the strongest excursion on the SEA curve inside the 0 to +5 years is used (which is often year 0 or +1).

| Type | Climate series | Season | Wine series | *p*-value |
|------|---------------|--------|-------------|-----------|
| Warmest | Luterbacher et al. (2004) temp. | DJF | Remich | 0.079 |
| Coldest | Luterbacher et al. (2004) temp. | DJF | Remich | **0.031 (–)** |
| Wettest | Pauling et al. (2006) precip. | DJF | Remich | **0.042 (–)** |
| Driest | Pauling et al. (2006) precip. | DJF | Remich | **0.012 (–)** |
| Warmest | Luterbacher et al. (2004) temp. | DJF | Grevenmacher | 0.483 |
| Coldest | Luterbacher et al. (2004) temp. | DJF | Grevenmacher | 0.054 |
| Wettest | Pauling et al. (2006) precip. | DJF | Grevenmacher | **0.007 (–)** |
| Driest | Pauling et al. (2006) precip. | DJF | Grevenmacher | 0.164 |
| Warmest | Luterbacher et al. (2004) temp. | MAM | Remich | **0.005 (+)** |
| Coldest | Luterbacher et al. (2004) temp. | MAM | Remich | **0.008 (–)** |
| Wettest | Pauling et al. (2006) precip. | MAM | Remich | **0.019 (–)** |
| Driest | Pauling et al. (2006) precip. | MAM | Remich | **0.010 (+)** |
| Warmest | Luterbacher et al. (2004) temp. | MAM | Grevenmacher | **0.003 (+)** |
| Coldest | Luterbacher et al. (2004) temp. | MAM | Grevenmacher | **0.022 (–)** |
| Wettest | Pauling et al. (2006) precip. | MAM | Grevenmacher | **0.002 (–)** |
| Driest | Pauling et al. (2006) precip. | MAM | Grevenmacher | **0.006 (+)** |
| Warmest | Luterbacher et al. (2004) temp. | JJA | Remich | 0.079 |
| Coldest | Luterbacher et al. (2004) temp. | JJA | Remich | **0.016 (–)** |
| Wettest | Pauling et al. (2006) precip. | JJA | Remich | 0.323 |
| Driest | Pauling et al. (2006) precip. | JJA | Remich | **0.006 (+)** |
| Warmest | Luterbacher et al. (2004) temp. | JJA | Grevenmacher | 0.083 |
| Coldest | Luterbacher et al. (2004) temp. | JJA | Grevenmacher | **0.035 (–)** |
| Wettest | Pauling et al. (2006) precip. | JJA | Grevenmacher | 0.090 |
| Driest | Pauling et al. (2006) precip. | JJA | Grevenmacher | 0.129 |
| Warmest | Luterbacher et al. (2004) temp. | SON | Remich | **0.020 (+)** |
| Coldest | Luterbacher et al. (2004) temp. | SON | Remich | 0.058 |
| Wettest | Pauling et al. (2006) precip. | SON | Remich | 0.051 |
| Driest | Pauling et al. (2006) precip. | SON | Remich | **0.049 (–)** |
| Warmest | Luterbacher et al. (2004) temp. | SON | Grevenmacher | **0.447 (+)** |
| Coldest | Luterbacher et al. (2004) temp. | SON | Grevenmacher | 0.181 |
| Wettest | Pauling et al. (2006) precip. | SON | Grevenmacher | 0.131 |
| Driest | Pauling et al. (2006) precip. | SON | Grevenmacher | 0.060 |

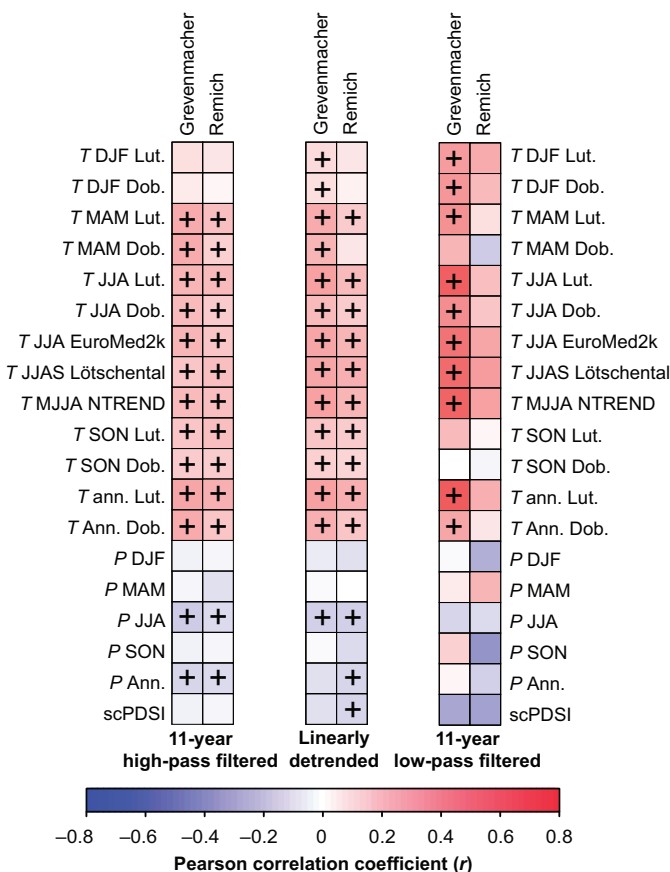

**Figure 4.** Correlations between the (palaeo)climate series listed in Table 2 (shown vertically) and wine production quantity data from Grevenmacher and Remich (shown horizontally). The three panels are for different temporal filtering. From left to right: 11-year high-pass filtered data, linearly detrended data, and 11-year low-pass filtered data (after linearly detrending). Correlations significant at $p = 0.05$ with the phase-scrambling test are marked with a plus sign (+).

= 0.46 and $r = 0.42$, respectively, using 11-year high-pass filtered data. No other seasons, neither for linearly detrended data nor high-pass filtered data, reveal any significant correlation between wine production and the De Bilt instrumental temperature series.

Using reconstructed temperature data covering the entire, or the majority, of the length of the Grevenmacher and Remich wine production series shows similar, but more significant, correlation patterns as found using the short De Bilt instrumental series. Mostly significant correlations are found between temperature and wine production quantities between both 11-year high-pass filtered and linearly detrended data (Fig. 4). These correlations are, in general, slightly stronger for Grevenmacher than for Remich. The most significant association is the positive relationship between spring, summer and autumn temperature, as well as for the annual mean, and wine production (i.e., warm = high wine production and *vice versa*). Correlations are weak for winter temperatures being entirely outside the growing season (Fig. 4). Significant positive correlations are also

found between (palaeo)climate series and the Grevenmacher series using 11-year low-pass filtered data. While some of the correlations are slightly stronger than those for high-pass filtered or linearly detrended data, the fewer degrees of freedom render a lower number of them significant. Notably, for Remich not a single correlation using 11-year low-pass filtered data is significant, although most of the correlations are positive and of comparable strength as those for high-pass filtered and linearly detrended data.

Negative relationships, in general less significant than those with temperature, are found in the hydroclimate between precipitation and soil moisture and wine production (i.e., wetter = lower wine production and *vice versa*) for both 11-year high-pass filtered data and linearly detrended data. Significant relationships with hydroclimate are mostly restricted to the summer season and to the annual mean. We note that while the correlation-based results provide the strongest signal for summer, the SEA-based results using extreme years instead show the strongest signal in spring. Nevertheless, both analyses reinforce the finding that warm and dry years and growing seasons benefited the wine production, while cold and wet years had an adverse effect. For hydroclimate, no significant correlations are found with either the Grevenmacher or Remich series using 11-year low-pass filtered (palaeo)data.

## 4 Discussion

### 4.1 The volcanic and climatic signals embedded in the wine production series

Our results are robust to choices regarding the exclusion of weaker volcanic forcing events as well as to the exclusion of individual particular strong volcanic forcing events. Although wine tithe data is zero, or close to it, in the first year following certain major volcanic events (1481, 1601, 1674, 1767 and 1784), this is not necessarily meaning that the vine harvest was zero or close to zero. From grain tithes, for example in the early modern Swedish Realm (Leijonhufvud, 2001; Huhtamaa et al., 2022), we know that sometimes no tithes were collected during years with extremely low harvests. Thus, zero wine tithe indicates very low vine harvests rather than necessarily no harvest at all. Furthermore, we cannot rule out the possibility that the respective error term is skewed towards a higher resistance to taxation in years of bad harvests. Such non-linearities are very difficult to assess and can similarly affect other types of records. Nevertheless, the long-term trend of the wine tithe data is more uncertain than the short-term (inter-annual to inter-decadal) variability. Since the SEA method focuses on a 5-year period before and after each event, it is not sensitive to potential long-term trends in the data. The compositing in SEA constitutes an averaging process that serves as a filter enhancing the high-frequency response signal of interest while minimising noise and also accounts for long-term drifts (see, e.g., Rao et al., 2019). We have also ensured that the response statistics to volcanic forcing are not driven by just a few major eruptions by both investigating the effects of including all volcanic forcing events and only those exceeding different threshold values. The results are robust also to using forcings less than a threshold. We have also tested the effects of excluding only the 1783 Laki eruption (not shown) and found that this volcanic forcing event is not dominating the results, although the forcing of Laki in the Northern Hemisphere is multiple times stronger than that of any other eruption.

Regarding the climate–wine production relationships, we can emphasise that we have tested both linear and non-linear correlation methods (that is Pearson correlation coefficient *versus* rank correlation methods such as Kendall and Spearman) although we only report the results of the first as they all give similar results. The similar results imply that the non-linearity that is indeed present in the wine data (which cannot go below zero) and various non-linearities present in palaeoclimate reconstructions have not prevented the linear Pearson correlation from rendering an accurate picture of how the series are correlated. The varying correlation strength between wine production and the different temperature reconstructions, even for the same season, is not surprising considering that the reconstructions have employed diverse input proxy data and different methods to combine the data and 'calibrate' the reconstructions against instrumental temperature measurements (see, for example, Christiansen and Ljungqvist, 2017; Anchukaitis and Smerdon, 2022).

Reduced wine production for up to several years following larger volcanic forcing events can, in part, be attributed to the fact that strong eruptions cause aerosols to linger for a longer period in the atmosphere, and induces feedback mechanisms in the climate system that prolongs cooling (e.g., sea-ice feedbacks, Miller et al., 2012; Toohey et al., 2016). In part, it can also be attributed to a biological memory effect, and cold and frost damage, of the vine stocks during years with very poor growth conditions (Meier et al., 2018; Pfister and Wanner, 2021). Note that while we do observe an immediate signal following larger volcanic events, the SEA results suggest that the maximum response is delayed. Biological memory effects are very well-documented for tree growth (Esper et al., 2015; Hartl-Meier et al., 2017). Especially, tree-ring width depends to some degree on growth conditions in the previous year and resulting climate reconstructions often show more auto-correlation than the instrumental target (Ljungqvist et al., 2020). As expected, auto-correlation values (AR1) reported for the time-series used in this study are higher, if biological archives are involved compared to documentary-based records (see Table 2). But while a limited immediate response to abrupt volcanic cooling has been reported for many tree-ring width records (Esper et al., 2015), as opposed to maximum latewood density (MXD) records (Hartl-Meier et al., 2017), this seems not to be the case for the wine production series. Thus, in this respect wine production shows a similar behaviour to MXD, potentially indicating that other physiological processes control fruit production than stem increment.

## 4.2 Further comparison between the volcanic effect on wine production and tree-ring growth

The wine production decline in Grevenmacher and Remich following volcanic forcing events contrasts to the weak volcanic response for Central Europe in the state-of-the-art gridded NTREND tree-ring-based temperature reconstruction (Anchukaitis et al., 2017), the mainly tree-ring-based updated EuroMed2k temperature reconstruction (Ljungqvist et al., 2019) as well as the MXD-based Lötschental temperature reconstruction (Büntgen et al., 2006). The NTREND reconstruction is for Europe primarily composed of MXD data from tree-rings, a temperature proxy commonly assumed to be little affected by biological memory (Anchukaitis et al., 2012; Esper et al., 2015). Despite remaining discussions regarding the precise quantification of volcanic signals in MXD-based reconstructions (Tingley et al., 2014; Edwards et al., 2022), there has been repeated evidence for significant cooling in MXD-based reconstructions following single events (Guillet et al., 2017; Hartl-Meier et al., 2017) and groups of eruptions at local (Rydval et al., 2017), regional (Esper et al., 2013b) and hemispheric (Wilson et al., 2016; Schneider et al., 2017) scales.

In contrast to a moderate volcanic cooling impact over Central Europe, NTREND and EuroMed2k reveal strong and widespread cooling in response to tropical eruptions over Scandinavia with a distinct transition between these two regions over the southern Baltic Sea region at around 55°N. However, NTREND and EuroMed2k do not contain tree-ring data in Europe between 50°N and 60°N making it difficult to determine the exact location of this transition. The gridded seasonal temperature reconstruction for Europe by Luterbacher et al. (2004), using predominantly documentary data, suggests significant summer cooling that extends as far south as the northern fringe of the Alpine arc (Fischer et al., 2007), which is better in agreement with the volcanic impact found here in wine production series from the Moselle Valley. The lack of climate-sensitive tree-ring records from Central Europe (north of the Alpine Arc) is also expressed in reduced values for explained variance at the Moselle grid-cell compared to the explained variance for, e.g., northern Fennoscandia.

Comparison between wine production declines and tree-ring growth declines is complicated by the limited number of major volcanic forcing events during the comparatively short period covered by the two wine production series. Numerous studies of temperature-sensitive tree-ring data, from northern Europe and elsewhere, have shown a maximum growth decline (growing season cooling) during the summer in the year following the volcanic forcing event (Esper et al., 2017). Such a distinct one year post-eruption cooling is most evident in MXD data, compared to TRW data, owing to the typically stronger correlation with temperature and the larger biological memory in TRW (Hartl-Meier et al., 2017). Thus, the post-volcanic decrease detected in this study in wine production quantities is very similar to the post-volcanic growth decline in MXD. Nevertheless, the most noteworthy feature in the wine production data is the strong and consistent response one year following the volcanic forcing event even for a low NH flux threshold ($> 0$ and $> 5$) that is absent in tree-ring based temperature reconstructions for Central Europe. At the mid-latitudes, where the wine production data diverges from the tree-ring based temperature reconstructions, no reconstructions show a significant summer temperature reduction in response to a low NH forcing flux threshold.

The Moselle Valley region is situated at the northern limit of viable viticulture areas in the same way as northern Scandinavia is situated close to the Arctic tree-line. Hence, both the Moselle Valley grapevines and the Scots pine trees in northern Scandinavia are thus very sensitive to temperature drops and presumably also to reduced sunlight after volcanic forcing events. The high climate sensitivity in both data types is clearly related to the marginally of the respective locations for grapevines and conifer trees. The higher climate sensitivity of the grapevines of Grevenmacher compared to Remich may also, rather than merely the small difference in latitude, be due to different soil conditions. We also suggest the explanation for the stronger, and more persistent, climate response in the wine production data than in the tree-ring data is due to physiological differences. In addition to the consistent signal one year following the volcanic forcing event, there is in the wine production data a 'late response' after big flux events (thresholds >15 and >20). This late signal is also present in the entirely tree-ring-based NTREND (Anchukaitis et al., 2017) and (weakly) in the Lötschental (Büntgen et al., 2006) temperature reconstructions (not shown). It is more difficult to come to a conclusion about this part of the signal because there are only few events falling into this category. While it would be interesting to study the climate (and volcanic) response on tree growth and wine production from the same locations, it would be challenging because temperature-sensitive and/or precipitation-sensitive tree-ring series rarely are available from vine growing regions with the exception of regions close to the Alps. Nevertheless, such studies could

provide insights into the decorrelation length of climate *versus* differences in the climate sensitivity of the vine growth and tree-ring growth.

## 5    Conclusions and outlook

We have systematically investigated the impacts of volcanic forcing events and climate variability on two of the longest wine production quantity records (spanning 1444–1786 with gaps) in Europe deriving from the Moselle Valley close to the northern limit of viticulture. We primarily used SEA analysis, which is particularly appropriate for episodic events. The statistical significance of the SEA was estimated with a bootstrap method. We also assessed the long-term relationship between climate and wine production using correlation analyses and calculating the significance with a conservative phase-scrambling test.

A strong negative impact of volcanic eruptions on wine production quantity was found in the year, and years, following a volcanic forcing event (subsection 3.1). This finding is robust under removal of weaker volcanic eruptions. Both SEA based on climate extremes and correlation analysis (11-year high- and low-pass filtered and linearly detrended) consistently show that cold and wet conditions are detrimental for wine production (subsection 3.3). Furthermore, following volcanic eruptions we see anomalously cold and wet conditions in the Moselle Valley region (subsection 3.2) which in the light of these results from subsection 3.3 indicates negative impacts on the wine production. This is exactly what we find in subsection 3.1. For both the volcanic impact on climate (subsection 3.2) and the impact of climate extremes on wine production (subsection 3.3), the largest effect is found in spring and summer. Thus, taken together, these findings present a consistent picture of the effect of volcanic eruptions on wine production and how it is mediated through climate.

We furthermore note that the detected volcanic signature in the Moselle Valley wine production is distinct and statistically significant (subsection 3.1), while that of tree-ring based temperature reconstructions for Central Europe is neither (subsection 3.2). Based on our findings we conclude that long series of annual wine production quantity contains very valuable biological and climatic information and we, thus, encourage further archival research to compile and publish additional wine production quantity series from across the viticulture regions of Europe.

*Code availability.*    We have used **IDL** and **R** (R Core Team, 2022) version 3.6.3 to program the analysis codes used in this work. In **R** we used the package 'corrplot' (Wei and Simko, 2021) to generate the correlation matrices. Data were read from mixed files using 'base' libraries for text files or 'openxlsx' (Schauberger and Walker, 2021) for Excel spreadsheet files. The superposed epoch analysis was performed with **R** code, available upon reasonable request.

*Data availability.*    The wine production data can be obtained from Appendix 1 in the article by Yante (1985), pp. 301–307. All used (palaeo)climate data are digitally available from the National Oceanic and NOAA Paleoclimatology/World Data Center for Paleoclimatology: https://www.ncei.noaa.gov/products/paleoclimatology. The monthly De Bilt temperature series (van Engelen and Geurts, 1985) are available

from the KNMI Climate Explorer (Trouet and Van Oldenborgh, 2013):

https://climexp.knmi.nl/getindices.cgi?WMO=KNMIData/labrijn&STATION=Tdebilt&TYPE=i

*Author contributions.* F.C.L. designed the study together with B.C. and P.T. F.C.L., B.C. and P.T performed data analyses. B.C. and P.T conducted the SEA and the phase-scrambling significance tests. L.S. especially contributed with comparisons between the volcanic signature in wine production data and tree-ring data. All authors interpreted the results and wrote the article together.

*Competing interests.* The authors have declared no conflicts of interest for this article.

*Acknowledgements.* F.C.L. acknowledges Visiting Researcher stays at the Institute of History, University of Bern, and at the Freiburg Institute for Advanced Studies (FRIAS), that allowed him time to work with this article. We express our appreciation to Asst. Prof. Michael Sigl, University of Bern, for lending us assistance regarding volcanic forcing datasets, and to Prof. em. Christian Pfister, University of Bern, for stimulating and insightful discussions about wine production data and grapevine phenology. The authors thank the four reviewers whose useful comments helped improve this article.

*Financial support.* F.C.L. was supported by the Marianne and Marcus Wallenberg Foundation (grant no. MMW 2022-0114) and the Swedish Research Council (Vetenskapsrådet, grant no. 2018-01272 and grant no. 2023-00605). He conducted the work with this article as a Pro Futura Scientia XIII Fellow funded by the Swedish Collegium for Advanced Study through Riksbankens Jubileumsfond. B.C. and P.T. were supported by the National Centre for Climate Research at the Danish Meteorological Institute. Open access publication funding for this article was provided by Stockholm University.

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
