# Peer review of "Strong volcanic-induced climatic shocks on historical Moselle wine production"

_Climate of the Past, 2024_

## Author Response (AR1)

**The Editor,** *Climate of the Past*

Fredrik Charpentier Ljungqvist
Professor of History, especially Historical Geography

Department of History
Stockholm University
SE-106 91 Stockholm
Sweden

E-mail: fredrik.c.l@historia.su.se
Mobile phone: +46706620728

[Figure]

Thursday, October 24, 2024

**Manuscript revision**

Dear editor,

Thank you very much for the new reviews of our manuscript entitled "Strong volcanic-induced climatic shocks on historical Moselle wine production" and for the opportunity to again revise the manuscript. We thank the reviewers for their comments and suggestions. We applied changes following the suggestions by the reviewers and added more information to improve the manuscript considering the recommendations. We hope that the revised version of the manuscript along with our answers to the reviewer's comments will make the article suitable for publication in *Climate of the Past*.

Thank you very much and we are looking forward to hearing from you soon.

We have implemented the following changes in response to the constructure comments by the reviewers:

(1) Regarding Fig. 4. This figure actually shows the correlations (colours) and the significance (+). The significance is calculated with a phase-scrambling method as described in the last paragraph of section 2.4. This is a stricter test than an ordinary t-test. Some of the confusion might come from the fact that the figure shows three panels (corresponding to different temporal filters). In each panel is shown 19×2 correlation coefficients; 19 is the number of available palaeoclimate series and 2 is the number of two wine series. The caption to Fig. 4 should now explain this better.

(2) The list of eruptions comes from a private communication but has been used by e.g. the IPCC. This is now mentioned in the text. l128.

(3) Note, that the years with actual volcanic eruptions should be selected with the same probability as any other year. Otherwise, a bias would be introduced. l177.

(4) The bootstrap approach using random years has been used for all SEA. For correlations we use the phase-scrambling method. This is described in section 2.4. These methods are both Monte Carlo approaches. The method is now, in the revised manuscript, mentioned also in the captions to Fig. 3 and Fig. 4. The method was already mentioned in Fig. 2 and Table 3.

(5) Fig. 3 now includes both the lag of the smallest p-value and the number of eruptions. The latter varies because of missing data in the palaeoclimate series.

(6) The 11-year filter is chosen to be consistent with the period of the SEA (5+5+1). This is now mentioned in l193. It is thus not chosen to match any solar cycle length.

(7) Units are now corrected throughout the manuscript.

(8) The word 'size' is now corrected to 'superposed signal strength'. l169.

(9) We have tested the sensitivity to the largest eruptions and included in the text and discuss this both in the method and in the results sections.

(10) All axis has been better labelled in Fig. 1 and Fig. 2.

(11) On lines 27–31 we now discuss and cite additional earlier studies about European climate–wine yield relationships.

(12) The work by Brönnimann and Krämer (2016) is now discussed on lines 59–61.

(13) Discussion section 4.1 is entirely rewritten now considering the input from the reviewers. We focus now on the effects of volcanic eruption size and uncertainties in the underlaying wine data.

(14) In Discussion section 4.2, we have added a new paragraph about the fact that the Moselle Valley is situated at the northern climatological limit for economic viable wine production just as the tree-ring series form northern Scandinavia comes from close to the Arctic tree-line. We emphasise that both the wine series and the tree-ring series from these locations are comparable in the sense that they derive from their temperature-limited edge.

On behalf of all authors,

Fredrik Charpentier Ljungqvist

---

## Author Response (AR2)

**Prof. Denis-Didier Rousseau,** *Climate of the Past*

Fredrik Charpentier Ljungqvist
Professor of History, especially Historical Geography

Department of History
Stockholm University
SE-106 91 Stockholm
Sweden

E-mail: fredrik.c.l@historia.su.se
Mobile phone: +46706620728

[Figure]

Monday, December 09, 2024

**Manuscript revision**

Dear Prof. Denis-Didier Rousseau,

Thank you very much for the new round of reviews of our manuscript entitled "Strong volcanic-induced climatic shocks on historical Moselle wine production" and for now accepting it pending corrections. We applied the changes suggested by the reviewers to Fig. 1 and Fig. 2 and hope these corrections now will make the article suitable for publication in *Climate of the Past*. We have also corrected a few minor typos in the text (commas, spelling etc.).

Regarding the comments by Reviewer #1, we already uploaded an answer to the Discussion by September 13, but for some technical error it was posted as four duplicates. When I asked for this to be corrected, all four responses (and not just the three duplicates) were removed. Our response to Reviewer #1 now reads:

"It is clear that the Reviewer envisions an entirely different type of article, with different set of methods, than the statistically based volcanic/climate–wine harvest study we have conducted. We have taken into account, and are very grateful for, the literature recommendations provided by the reviewer. But since the reviewer – as opposed to the other three reviewers – appears to wish for an entirely other type of study, we have not been able to otherwise adhere to the review."

On behalf of all authors,

Fredrik Charpentier Ljungqvist